# Short Peptides of Innate Immunity Protein Tag7 (PGLYRP1) Selectively Induce Inhibition or Activation of Tumor Cell Death via TNF Receptor

**DOI:** 10.3390/ijms241411363

**Published:** 2023-07-12

**Authors:** Daria M. Yurkina, Tatiana N. Sharapova, Elena A. Romanova, Denis V. Yashin, Lidia P. Sashchenko

**Affiliations:** Institute of Gene Biology (RAS), Moscow 119334, Russia; yrkina121@gmail.com (D.M.Y.); sharapovatat.nik@gmail.com (T.N.S.); elrom4@rambler.ru (E.A.R.); sashchenko@genebiology.ru (L.P.S.)

**Keywords:** Tag7, Hsp70, TNFR1, signal transduction, cytotoxity, short peptides

## Abstract

In this study, we have found two peptides of Tag7 (PGLYRP1) protein-17.1A (HRDVQRT) and 17.1B (RSNYVLKG), that have different affinities to the TNFR1 receptor and the Hsp70 protein. Peptide 17.1A is able to inhibit signal transduction through the TNFR1 receptor, and peptide 17.1B can activate this receptor in a complex with Hsp70. Thus, it is possible to modulate the activity of the TNFR1 receptor and further perform its specific inhibition or activation in the treatment of various autoimmune or oncological diseases.

## 1. Introduction

The identification of new ligands of specific receptors responsible for the regulation of immune responses, and the uncovering of these proteins’ mechanisms of action, is essential for understanding the functioning of the immune system.

The TNF cytokine receptor TNFR1 protein plays an important role in immune response and autoimmune diseases [1,2]. It plays a major role in systemic inflammation [3], cancer development [4], septic shock [5] and autoimmune arthritis [6], as well as tuberculosis treatment [7] and immune system development [8]. We have found that the Tag7 protein (PGLYRP1) is a ligand of the TNFR1 proinflammatory receptor present on the surface of many cells, including immune and tumor cells [9,10]. Via binding to TNFR1, Tag7 inhibits the death of tumor cells under the action of a specific ligand of this TNF receptor or DNA-hydrolyzing autoantibodies [10,11]. We also showed that Tag7 forms a stable, equimolar cytotoxic complex with the main heat shock protein, Hsp70, which induces an apoptotic and necroptotic pathway of tumor cell death when interacting with TNFR1 [10].

The study of the mechanisms of tumor cell death under the action of cytotoxic T lymphocytes allowed us to establish that Tag7 exposed on the cell surface of CD4+-T lymphocytes can interact with Hsp70 on the surface of a tumor cell and ensure the binding of a lymphocyte to a target cell with subsequent induction of cell death due to the interaction of the FasL lymphocyte with the Fas receptor of a tumor cell [12].

One of the approaches to understanding the mechanisms of action of proteins in the regulation of the immune response, as well as the use of these proteins in immunotherapy, is to identify biologically active epitopes of regulatory proteins interacting with receptors. Also, peptide fragments can simulate the functional activity of a full-sized protein. By binding to the receptor, they can both inhibit and activate its functions. Such functional peptides can be used as medications [13].

Recently, we have isolated peptide 17.1, located in the C-terminal region of the polypeptide chain of the Tag7 protein, which has the bifunctional activity inherent in a full-sized protein. Like Tag7, peptide 17.1 bound to TNFR1 and inhibited the cell death processes induced by other ligands of this receptor [14]. In combination with Hsp70, this peptide initiated apoptotic and necroptotic processes in the tumor cell [14]. It has also been shown that the shortened fragment of this peptide, peptide 17.1A, also inhibits the activation of TNFR1 under the action of other ligands of this receptor [15]. Both peptides showed a protective effect in the development of CFA (complete Freund’s adjuvant)-dependent autoimmune arthritis in a mouse model [14,15].

The purpose of this study was to expand the understanding of the functional activity of peptide 17.1, as follows: (1) to divide the activity of bifunctional peptide 17.1 to identify its shortened fragments interacting with the TNFR1 receptor or with the Hsp70 protein and (2) to find out the possibility of the binding of this peptide and its shortened fragments with the TNFR1 receptor on the surface of tumor cells and their effect on cytotoxic activity CD4+-T lymphocytes.

## 2. Results

### 2.1. Peptides 17.1A and 17.1B Bind to TNFR1 on the Cell Surface

Previously, we found that peptide 17.1 (a.a. 141–157) is located in the C-terminal section of the polypeptide chain of the Tag 7 protein (PGLYRP1) [14]. The 3D structure of this protein is shown in Appendix A, and the peptides used in this work are marked on it. It can be seen that the 3D structure of peptide 17.1 is heterogeneous. The *N*-terminal fragment of this peptide (a.a. 141–149) is not structured. The fragment corresponding to the next section of the polypeptide chain (a.a. 150–156) represents the structure of the β-sheet. To separate the activities of bifunctional peptide 17.1, we synthesized peptides: 17.1B, corresponding to the amino acid sequence of the *N*-terminal fragment (RSNYVLKG) and 17.1A, corresponding to the amino acid sequence of the β-leaf (HRDVQRT).

First, the ability of these peptides to bind to the TNFR1 receptor in solution was determined. For this purpose, affinity chromatography was performed using a column with a soluble TNFR1 receptor domain (sTNFR1) immobilized on CN-Br-Sepharose (Figure 1a). Earlier, we demonstrated [15], and here, we confirmed that the peptide 17.1A binds to sTNFR1. The interaction of peptide 17.1B with the soluble TNFR1 site could not be detected. It is possible that the 17.1B–TNFR1 complex dissociates rapidly due to the peptide’s weak affinity for this receptor.

However, as can be seen from the results in Figure 1b, both peptides interact with the TNFR1 receptor exposed on the cell surface. After incubation of the cells with biotinylated peptides in the presence of the BS^3^ cross-linking reagent, the cell lysate was purified by magnetic separation with Streptavidin-coated magnetic bodies. The bound material was separated and analyzed using SDS-PAGE and Western blot and was detected using specific antibodies to TNFR1. As a result, two complexes of the peptides 17.1A and 17.1B with TNFR1 with a molecular weight of 52 kDa can be seen (Figure 1b). The detection of the 17.1B-TNFR1 complex was weaker, possibly due to the lower affinity of this peptide to the receptor.

### 2.2. Peptide 17.1A Inhibits TNFR1-Dependent Cytotoxic Activity

TNFR1 is known to induce alternative cytotoxic processes that develop at different time intervals (apoptosis 3 h after ligand addition, necroptosis 20 h later). We have also shown that the cytotoxic Tag7-Hsp 70 complex and DNA-hydrolyzing autoantibodies can be ligands of TNFR1 [10,11]. Next, we investigated the ability of synthesized peptides to compete with TNFR1 ligands and inhibit TNFR1-induced cytotoxicity. The results are shown in Figure 2. Previously, we showed that peptide 17.1A inhibits the death of mouse cells of the L-929 line under the action of TNF, the Tag7-Hsp 70 complex and DNA-hydrolyzing autoantibodies. Here, we confirmed these results by showing that peptide 17.1A prevents the development of both apoptosis and necroptosis under the action of these ligands when human leukemia cells of the HL 60 line are used as target cells (Figure 2a,b).

The results in Figure 2c–f indicate that, unlike peptide 17.1A, peptide 17.1B did not inhibit the development of either apoptosis or necroptosis under the action of three TNFR1 ligands. The cytotoxic activity did not decrease when using both mouse fibroblasts and human leukemia cells as target cells.

The dependence of the inhibition on the concentration of peptide 17.1A was dose-dependent; half of the maximum inhibition (I_50_) was 1 nM. Peptide 17.1B showed a weak inhibitory effect only at a concentration of 100 nM (Figure 3).

Thus, peptide 17.1A fully reproduces the inhibitory activity of peptide 17.1 and the full-sized Tag7 protein. Peptide 17.1B appears to have a very low affinity for TNFR1 and is easily displaced by ligands of this receptor.

### 2.3. Peptide 17.1B Induces Cytotoxicity in Complex with Hsp 70

Next, we found out which fragments of peptide 17.1 are responsible for its other function—the ability to kill tumor cells in complex with Hsp70.

The interaction of both peptides with Hsp70 was also studied using affinity chromatography. The peptides bound to Hsp70 immobilized on Br-CN Sepharose were analyzed using Tricine–SDS-PAGE and detected by Coomassie staining (Figure 4a). It can be seen that both peptide fragments have an affinity for Hsp70. After staining, a clear band corresponding to the mobility of the peptide 17.1B and a weaker band corresponding to the mobility of the peptide 17.1A are detected.

The stability of Hsp70 complexes with the studied peptides and their ability to induce the programmed cell death of tumor cells were investigated. As can be seen from the results in Figure 4b, the cytotoxic activity of the 17.1B-Hsp70 complex is comparable to the cytotoxic activity of the 17.1-Hsp70 and Tag7-Hsp70 complexes. Cytotoxic activity of the 17.1A-Hsp70 complex was not detected. The dependence of cytotoxic activity on the concentration of the 17.1B-Hsp70 complex was dose-dependent; maximum cytotoxicity was achieved at a concentration of 10 nM (Figure 4c). The 17.1A-Hsp70 complex also showed a weak cytotoxic effect at a concentration of 100 nM. These results suggest a lower affinity of peptide 17.1A to the Hsp70.

### 2.4. Peptide 17.1B Inhibits the Cytotoxic Activity of CD4+-T Lymphocytes

As mentioned above, we have shown the participation of Tag7 exposed on the cell membrane of cytotoxic CD4+-T lymphocytes in the cytolysis of Hsp70-positive tumor cells by these lymphocytes. This protein ensured the binding of the lymphocyte to the target cell due to the formation of the intercellular Tag7-Hsp70 complex.

Here, we found out whether the peptides 17.1A and 17.1B bind to Hsp70 on the membrane of a tumor cell and whether this binding also prevents the interaction of a cytotoxic lymphocyte with this cell. Biotinylated peptides were incubated with cells in the presence of the BS^3^ cross-linking reagent.

The cell lysates were purified by magnetic separation with Streptavidin-conjugated magnetic beads. The bound material was analyzed using SDS-PAGE followed by a Western blot using specific antibodies to Hsp70. This made it possible to identify two complexes of the biotinylated peptides 17.1A and 17.1B with Hsp70 with a mass of 70 kDa (Figure 5a). Thus, both fragments of peptide 17.1 can bind to Hsp70.

Furthermore, we investigated how this binding affects the induction of cell death by cytotoxic CD4+-T lymphocytes. Tumor cells of the K562 line were preincubated with the peptides 17.1, 17.1A and 17.1B, and then cytotoxic lymphocytes were added and the cytotoxic activity was determined. It can be seen that peptides 17.1 and 17.1B completely block the cytotoxic effect of lymphocytes, whereas peptide 17.1A has no effect on cell death (Figure 5b). Thus, although all three peptides can bind to Hsp70 on the cell surface, only peptide 17.1 and its fragment 17.1B have inhibitory activity. Apparently, peptide 17.1A has a low affinity for Hsp70 and forms an unstable complex with it.

### 2.5. Determination of the Binding Constants of Peptides 17.1, 17.1A and 17.1B with TNFR1 and Hsp70

Protein–protein interactions of sTNFR1 and Hsp70 with the peptides 17.1, 17.1A and 17.1B and the Tag7 protein were quantified using microscale thermophoresis (MST). This biophysical method is based on the movement of molecules in a temperature gradient, which strongly depends on the charge, the hydration shell and the size of the moving molecules. At least one of these qualities usually changes during the formation of the complex. Therefore, thermophoresis provides a sensitive and reliable method for analyzing and quantifying protein–protein interactions [16,17,18]. The addition of peptides to labeled TNFR1 shows clear changes in thermophoresis, and the thermophoretic signals obtained with increasing concentrations follow a clear binding curve (Figure 6). From these binding curves, the apparent dissociation constants (Kd) presented in Table 1 were obtained. The obtained constants indicate a highly specific interaction of Tag7, 17.1 and 17.1A with sTNFR1. Similarly, clear changes in the thermophoretic signal were observed when Tag7, 17.1 and 17.1B were added to the labeled Hsp70. The corresponding binding curves showed nanomolar values of apparent Kd. However, the addition of peptide 17.1B to labeled sTNFR1 showed a significantly lower apparent dissociation constant (Kd) equal to 6690(120) nM, which suggests a lower affinity of these molecules and well describes the inability of peptide 17.1B to compete with TNFR1 ligands for binding to this receptor. The same pattern was observed when peptide 17.1A was added to labeled Hsp70. The apparent dissociation constant (Kd) of 17.1A-Hsp70 was determined as 222(15) nM, which is also significantly lower than that in the 17.1B-Hsp70 complex. It is obvious that peptide 17.1A cannot form a strong complex with Hsp70, and this may explain its inability in combination with Hsp70 to cause TNFR1 receptor activation.

## 3. Discussion

In summarizing the above results, it should be emphasized that the paper expands the understanding of the mechanisms of regulatory action of the innate immunity protein Tag7 (PGLYRP1). For the first time, Tag7 epitopes responsible for the opposite functions of this protein were identified: the inhibition of the TNF–TNFR1 interaction and the induction of programmed tumor cell death in combination with the Hsp70 protein via the TNFR1 receptor.

The TNF cytokine plays a key role in the development of the immune system and the proinflammatory immune response [19]. It can also cause the death of tumor cells [20]. TNF interacts with the TNFR1 receptor, which is present on the surface of most cells, including tumor cells [21]. The interaction of TNF with TNFR1 can induce intracellular signaling. Depending on the activity of the enzymes associated with the intracellular domain of this receptor, this may be gene activation under the control of transcription factor NFkB or programmed cell death [22]. NFkB-dependent activation regulates immune cell proliferation and the proinflammatory immune response [23]. However, excessive TNF activity leads to the emergence of many autoimmune diseases, and the regulation of the activity of this cytokine is an urgent task of modern immunology [24]. It is known that antibodies to this cytokine and the soluble extracellular domain of its receptor (sTNFR1) are successfully used to suppress TNF activity [25].

We have previously shown that DNA-hydrolyzing autoantibodies and the innate immunity protein Tag7 can interact with TNFR1 and act as its ligands [10,11].

Tag7 (PGLYPR1) is a multifunctional protein involved in the regulation of many immune processes [26]. Its antibacterial activity is known [27]. It has been shown that in synergy with the Toll receptor, it participates in antibacterial protection in insects. In mammals, it also prevents the development of bacterial infection [27]. It has recently been established that PGLYPR1 is a ligand of the innate immunity receptor TREM1, whose main function is the secretion of proinflammatory cytokines [28]. In this case, Tag7 acts as an inducer of inflammatory processes. It promotes the activation of proinflammatory cytokine genes, including TNF [29]. The binding of Tag7 to TNFR1 prevents the interaction of this receptor with its TNF ligand, and here, Tag7 can be considered an anti-inflammatory cytokine [10].

The ability of Tag7 to form functional complexes with other proteins expands the range of its functions. The Tag7 complex with the Mts1 protein belonging to the S100 family of proteins induces lymphocyte chemotaxis [30]. When Tag7 interacts with Hsp70, a cytotoxic complex is created [31]. Tag7 in this complex retains the ability to interact with TNFR1, but its function is reversed. It does not inhibit the cytotoxic effect of TNF, but on the contrary, it acts like a TNF cytokine, inducing the programmed death of tumor cells.

A serious direction of modern biological science is the identification of peptide fragments of multifunctional proteins that perform one specific function. Such studies provide approaches to the detailed decoding of the mechanism of action of these proteins and the creation of new drugs [22]. As mentioned above, we have localized the Tag7 site required to perform its two functions [14]. Peptide 17.1 was synthesized with an amino acid sequence corresponding to this fragment, simulating both functional activities of a full-sized protein. When added to TNFR1, this peptide inhibited the cytotoxic effect of its ligands [14]. In complex with Hsp70, peptide 17.1 induced the death of tumor cells. This peptide showed anti-inflammatory activity. With the development of autoimmune arthritis, it significantly reduced the destruction of cartilage and bone tissue and had an effect on the secretion of a wide range of cytokines and chemokines [14].

Here, we managed to separate the functions of the bifunctional peptide 17.1. It was found that the *N*-terminal fragment of this peptide, the 8-membered peptide 17.1A, performs only one function. This peptide proved to be an effective inhibitor of the cytotoxic activity of TNF and autoantibodies, but unlike peptide 17.1, it did not induce cell death in complex with Hsp70 at physiological conditions. A quantitative assessment of the interaction of peptide 17.1A with both proteins indicates its high affinity for the TNFR1 receptor, almost the same as that of Tag7 and peptide 17.1, and significantly lower affinity for the Hsp70 protein.

The *C*-terminal fragment of the peptide 17.1–7-membered peptide 17.1B also performs only one function under physiological conditions. It had a low affinity for the TNFR1 receptor and did not inhibit the cytotoxicity of TNF and other ligands of this receptor. At the same time, like Tag7 and peptide 17.1, it also bound to Hsp70 with higher affinity and formed a stable cytotoxic complex capable of killing tumor cells.

The high affinity of peptides 17.1 and 17.1B to Hsp70 allows them to form stable complexes (17.1-Hsp70 and 17.1B-Hsp70) on the surface of Hsp70+ tumor cells and prevent CD4 + CD25 + lymphocytes from interacting with these cells, blocking the cytotoxic activity of these lymphocytes. Peptide 17.1A, which has a low affinity for Hsp70, easily dissociates from the cell surface and does not inhibit the death of tumor cells under the action of these lymphocytes.

Thus, fragments of Tag7 responsible for the implementation of its anti-inflammatory and antitumor functions have been identified. The results obtained can be used in autoimmune and antitumor therapy.

## 4. Materials and Methods

### 4.1. Cell Cultivation and Sorting

K562 and HL-60 cells were cultured in RPMI-1640 with 2 mm L-glutamine and 10% FCS (Invitrogen, Carlsbad, CA, USA). L929 cells were cultured in DMEM with 2 mm L-glutamine and 10% FCS (Invitrogen, Carlsbad, CA, USA). These cell lines were obtained from the cell line collection of the N. N. Blokhin National Medical Research Center of Oncology of the Ministry of Health of Russia. Human peripheral blood mononuclear cells (PBMC) were isolated from the total pool of leukocytes of healthy donors, as described in ref. [31], and cultured at a density of 4 × 10^6^ cells/mL in the same medium with Tag7 (1 nM) for 6 days.

### 4.2. Proteins and Antibodies

Recombinant Tag7, sTNFR1 and Hsp70 were obtained as described in [10].

### 4.3. Peptides

The peptides 17.1, 17.1A and 17.1B were synthesized on an automated peptide synthesizer in accordance with the Fmoc strategy, and HATU (azabenzotriazole tetramethyluronium hexafluorophosphate) was used as a binding agent. Amino acids were taken in an eight-fold excess; the condensation of each amino acid was carried out for 30 min. *C*-terminal amino acids were attached to the activated resin in the presence of DIPEA for 2 h. After synthesis, the protected peptidyl polymer was washed with diethyl ether, dried and treated with a mixture of TFA/DTT/H_2_O/TIS (150/4/3/0.5 wt. %) (15 mL of the mixture per g of peptidyl polymer) for 2 h. The solution was filtered; the untreated peptide was precipitated with a tenfold volume of diethyl ether and kept at a temperature of 4 °C for 8 h. The precipitated peptide was centrifuged, washed three times with diethyl ether and dried under vacuum. The untreated peptide was purified by HPLC on a YMC Actus Triart C18 10u 30 × 150 mm column in a gradient of 5–55% acetonitrile and lyophilized [32].

### 4.4. Affinity Chromatography, Immunoadsorption and Immunoblotting

A column with CNBr-activated Sepharose 4B (GE Healthcare, Chicago, IL, USA) conjugated with sTNFR1 or Hsp70 was prepared as described in accordance with the manufacturer’s protocol. The peptides 17.1, 17.1A and 17.1B were biotinylated according to the protocol from [33]. The biotinylated peptides were loaded into a column with sTNFR1 or Hsp70-Sepharose. The excess peptide was washed out of the column using PBS containing 0.5 M NaCl and only PBS. The peptide was eluted with 0.25 M triethylamine (TEA), pH 12. The eluted material was resolved using Tricine–SDS-PAGE and applied to a nitrocellulose membrane [34]. HRP-conjugated Streptavidin (GE Healthcare, Chicago, IL, USA; 1:15,000; 1 h) was used for detection. The results were visualized using the ECL Plus kit (GE Healthcare, Chicago, IL, USA) in accordance with the manufacturer’s protocol. The chemiluminescence was recorded using iBright (Thermo Fisher Scientific, Boston, MA, USA). L929 or K562 cells (10^8^ cells) were incubated with the biotinylated peptides (1 nM) in the presence of BS^3^ (Thermo Fisher Scientific, Boston, MA, USA), lysed in RIPA buffer (Sigma-Aldrich, St. Louis, MO, USA) and purified using Dynabeads (M-280 sheep anti-rabbit IgG; Dynal Biotech ASA, Oslo, Norway), conjugated with Streptavidin in accordance with the manufacturer’s protocol. This material was resolved into 10% SDS-PAGE, followed by Western blotting using antibodies against TNFR1 or against Hsp70 (Abcam, Cambridge, UK).

### 4.5. Cytotoxicity Assays

For the cytotoxicity tests, target cells were cultured in 96-well plates (6 × 10^4^ cells per well) and mixed with lymphocytes at a ratio of 1:20. The cytotoxicity was measured after 24 h of incubation. The inhibition test was performed with peptides 17.1, 17.1A and 17.1B at a concentration of 1 nM. The cytotoxic activity of the lymphocytes was determined using the Cytotox 96 analysis kit (Promega, Madison, WI, USA) in accordance with the manufacturer’s protocol.

### 4.6. Microscale Thermophoresis

The purified sTNFR1 and Hsp70 were fluorescently labeled using the Alexa Fluor 633 or Alexa Fluor 488 staining kit (Eugene, OR, USA) in accordance with the manufacturer’s instructions. sTNFR1 and Hsp70 (200 nM) were incubated for 20 min with each compound in the dark at room temperature at 16 different concentrations obtained by sequential dilution, starting from the highest soluble concentration. The samples were loaded into glass capillaries (Monolith NT Capillaries) and analyzed by thermophoresis using Nano-Temperature Monolith NT 115 apparatus (IR laser power 10%). The signal quality was monitored by a NanoTemper Monolith device to detect possible autofluorescence of the ligand, deposition, aggregation or ligand-induced changes in the photobleaching rate. The experiments were carried out in triplicate and processed using affinity analysis software (MO Control v.1.6.1, Nano-Temper).

### 4.7. Statistical Analysis

The data were analyzed using Statistica 6.1 (StatSoft^®^) software. The Shapiro–Wilk test was used to confirm the normality of the distribution of the data. The results are presented as an average value ± SD. Statistically significant differences were determined using the *t*-test. The value of *p* < 0.05 was considered statistically significant.

## 5. Conclusions

We show that two peptides, 17.1A and 17.1B, have different affinities to the TNFR1 receptor and the Hsp70 protein, and each of them is able to perform only one function at physiological conditions: peptide 17.1A is only able to inhibit signal transduction through the TNFR1 receptor, and peptide 17.1B can only activate this receptor in complex with Hsp70.

## Figures and Tables

**Figure 1 ijms-24-11363-f001:**
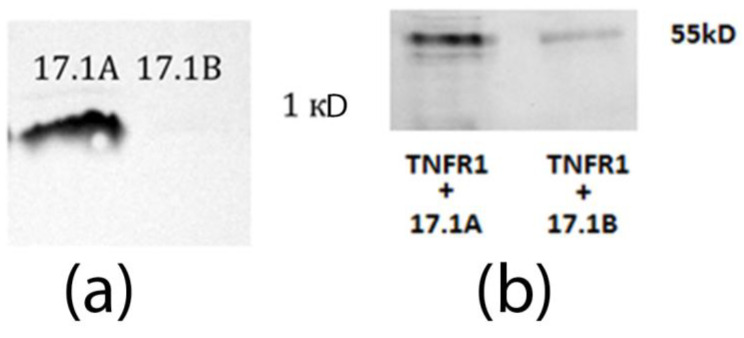
(**a**) The peptides were biotinylated and applied to a column containing TNFR1-conjugated Sepharose. The bound material was eluted with triethylamine and resolved using Tricine–SDS-PAGE and visualized using Streptavidin conjugated with peroxidase and ECL Plus kit (GE Healthcare, Chicago, IL, USA). (**b**) The peptides were biotinylated and added to cells of the L929 line, followed by incubation for 1 h. Next, the peptides were cross-linked with TNFR1 on the cell surface with BS^3^ reagent (Thermo Fisher Scientific, Boston, MA, USA). After stopping the reaction, the cells were lysed in RIP-A (Sigma-Aldrich, St. Louis, MO, USA) buffer in the presence of protease inhibitors. The lysate was purified by magnetic separation on M280 Streptavidin Beads. The obtained fractions containing 17.1 A-TNFR1 and 17.1 B-TNFR1 were analyzed using 12% SDS-PAGE electrophoresis and Western blot. Complexes were detected using primary polyclonal antibodies abTNFR1 and secondary antibodies anti-rabbit peroxidase-linked (ECL).

**Figure 2 ijms-24-11363-f002:**
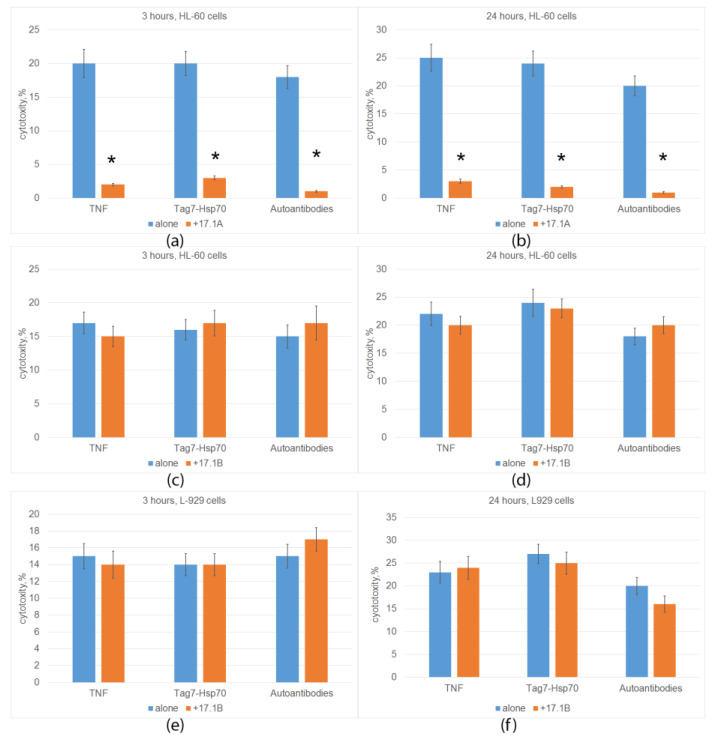
Cells (HL-60 (**a**–**d**) or L-929 (**e**,**f**)) were preincubated with peptide 17.1A (**a**,**b**) or 17.1B (**c**–**f**) for 30 min, and then a cytotoxic agent (TNF, Tag7-Hsp70 complex or autoantibodies) was added and cytotoxic activity was then determined after 3 (**a**,**c**,**e**) or 24 h (**b**,**d**,**f**) of cell incubation. *n* = 5 for each group). (* *p*-value < 0.05).

**Figure 3 ijms-24-11363-f003:**
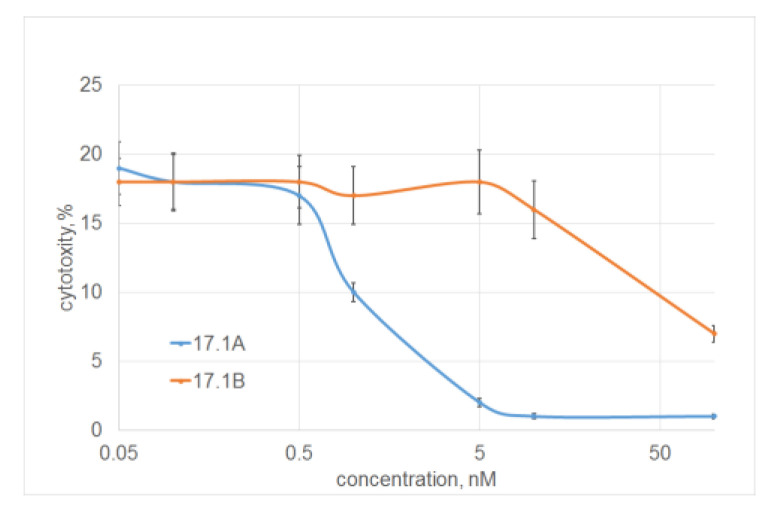
Dependence of inhibition of TNF cytotoxic activity on peptide 17.1 and 17.1B concentrations. (L929 cells, 24 h of incubation, peptides were added 30 min before incubation with TNF (1 nM)) *n* = 5 for each point.

**Figure 4 ijms-24-11363-f004:**
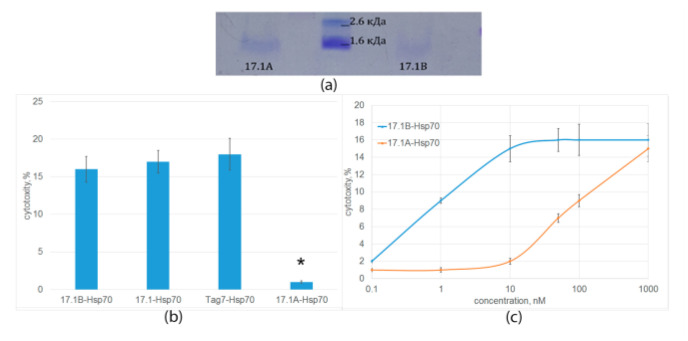
(**a**) Peptides 17.1A and 17.1 B were incubated with Hsp70 in solution, coimmunoprecipitated with anti-Hsp70 Sepharose and the resulting material was resolved using Tricine–SDS-PAGE. (**b**) Cytotoxic activity of Hsp70 protein complexes with full-size Tag7, peptides 17.1, 17.1A and 17.1B. (10 nM) (* *p*-value < 0.05). (**c**) Concentration dependence of cytotoxic activity of 17.1-Hsp70 and 17.1B-Hsp70 complexes (L929 cells, 24 h of incubation) *n* = 5 for each point.

**Figure 5 ijms-24-11363-f005:**
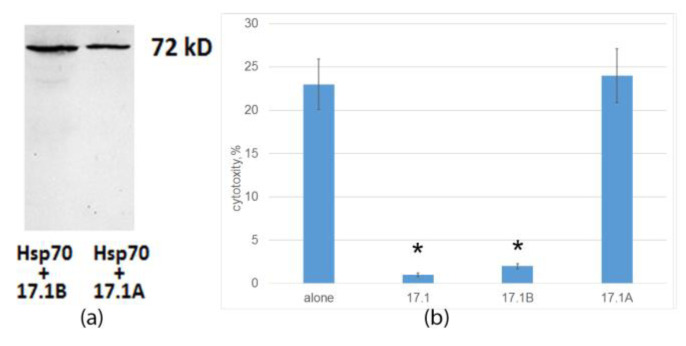
(**a**) The peptides were biotinylated and added to cells of the K562 line, followed by incubation for 1 h. Next, the peptides were cross-linked with Hsp70 on the cell surface with BS^3^ reagent (Sigma-Aldrich, St. Louis, MO, USA). After stopping the reaction, the cells were lysed in RIP-A (Sigma-Aldrich, St. Louis, MO, USA) buffer in the presence of protease inhibitors. The lysate was purified by magnetic separation on M280 Streptavidin Beads. The obtained fractions containing 17.1A-Hsp70 and 17.1B-Hsp70 were analyzed using 12% SDS-PAGE electrophoresis and Western blot. Complexes were detected using primary antibodies abHsp70 poly-AT and secondary antibodies anti-rabbit peroxidase-linked (ECL). (**b**) Tumor cells of the K562 line were preincubated with peptides 17.1, 17.1A and 17.1B for 30 min., and then cytotoxic lymphocytes were added and cytotoxic activity was determined (24 h incubation), *n* = 5 for each point, (* *p*-value < 0.05).

**Figure 6 ijms-24-11363-f006:**
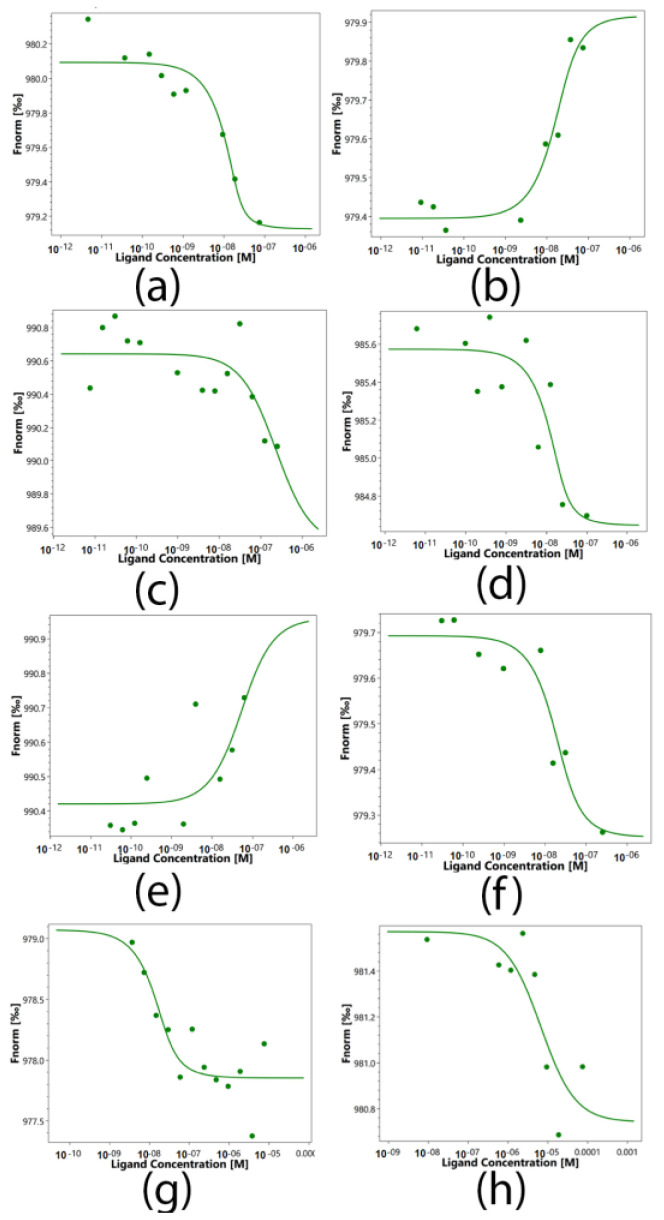
Microscale thermophoresis data for the interaction of Tag7 (**a**,**e**), peptides 17.1 (**b**,**f**), 17.1 (**c**,**g**) and 17.1 B (**d**,**h**) with proteins Hsp70 (**a**–**d**) and sTNFR1 (**e**–**h**). Each experiment was performed in triplicate, and the most common data are shown.

**Table 1 ijms-24-11363-t001:** Apparent dissociation constants obtained from microscale thermophoresis.

Ligands	K_d_, nM
Tag7-Hsp70	1.73 ± 0.3
17.1-Hsp70	7.16 ± 1
17.1A-Hsp70	222 ± 15
17.1B-Hsp70	3.15 ± 0.7
Tag7-TNFR1	43.2 ± 5
17.1-TNFR1	8.84 ± 1
17.1A-TNFR1	5.44 ± 1
17.1B-TNFR1	6690 ± 120

## Data Availability

Not applicable.

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
