# Peer review of "Short Peptides of Innate Immunity Protein Tag7 (PGLYRP1) Selectively Induce Inhibition or Activation of Tumor Cell Death via TNF Receptor"

_ijms, 2023, doi:10.3390/ijms241411363_

Round 1
Reviewer 1 Report
The article is devoted to an urgent scientific and practical topic – the development of peptide drugs for the treatment of oncological and autoimmune diseases. To achieve this task, the authors apply a wide range of methods of molecular biology, which confirms the representativeness of the results. In addition to small questions and comments, there is one significant remark. On graphs (figures 2a, b; 4b, 5b), the error of the average is greater than the average value of the measured value. In our opinion, especially when using the t-test, this is unacceptable. It raises doubts about the reliability of obtained results. At the same time, there is no detailed description of statistical data processing. We really hope that authors would be so kind to eliminate this remark, which will allow the article to be accepted for publication in IJMS.
1. What criterion was used to confirm that the data belongs to the normal distribution?
2. In Figures 2 a,b (autoantibodies columns); 4b, 5b (peptide 17.1), the spread of values is greater than the average. It is necessary to carefully check the statistical analysis of the data. Perhaps it needs another method of data analysis? In the presented form, the results do not look reliable.
3. Where were the K562 and HL-60 cell lines obtained from? Why these lines were chosen for the study?
4. It is necessary to give the structures of the studied peptides in the abstract and in the first mention in the text.
Author Response
1-2. We are very grateful to the reviewer for the careful reading of our work and the comments made. We deeply apologize, indeed, the confidence intervals are incorrectly displayed in the figures. We have corrected the presented data and confidence intervals. We have made changes to Figures 2-5 to correct this mistake.
- We have added information about the sources of K562 and HL-60 cells to the Materials and Methods chapter. K562 cells were used to determine the cytotoxic activity of lymphocytes, and are good targets for lymphocytes, as shown by our previous experiments. (references 12 and 30 in the manuscript) HL-60 cells were tested for sensitivity to TNFR1-dependent lysis, and they showed high sensitivity. We decided that the results of cytotoxic activity on cells of human origin would complement our results obtained on mouse L929 cells.
- We have added the sequence of peptides used in the work to the abstract and Introduction.
Reviewer 2 Report
Manuscript Title: Short peptides of innate immunity protein Tag7 (PGLYRP1) selectively induce inhibition or activation of tumor cell death via TNF receptor
Manuscript Number: ijms-2491003
Article Type: Article
Comments:
The manuscript “Short peptides of innate immunity protein Tag7 (PGLYRP1) selectively induce inhibition or activation of tumor cell death via TNF receptor” by Denis V. Yashin and co-workers reported innate immunity protein Tag7 (PGLYRP1) selectively induce tumor cell death through TNF receptor. In this study, the authors show that two peptides derived from 17.1-17.1A and 17.1B have different affinities to the TNFR1 receptor and the HSP70 protein.
I thoroughly read the whole manuscript. The authors drafted the manuscript with good supporting data execution, interpretation, and reference citations. I have found some grammatical errors and do not see the proper spacing between the words and errors in reference citations. Please correct the grammatical mistakes carefully. However, the authors provided some valuable research that could be interesting. After careful reading, I am considering this manuscript needs a minor revision. Please re-submit the manuscript with the improved and revised version.
Modest items that need to be addressed include:
1. The abstract description is not convincing to me and please revise it. Please remove the first two sentences in the abstract and add them to the introduction part. The abstract should be precise and include your key observations and findings in a systematic way,
2. Manuscript English grammar needs to be corrected.
3. The introduction part needs to be revised with supporting references.
4. I dint find any schemes, procedures, and analytical supporting data for the reported peptides in this manuscript except cite your reference. It would be good to add the synthesis part to this draft.
5. It would be good if you could add some more references or reviews in the introduction part.
Manuscript English grammar needs to be corrected
Author Response
- We thank the reviewer for the careful reading of our work and the comments made. According to the comments made, we have changed the abstract of the work.
- We have made changes to the text to correct the English language.
- We have made changes to the Introduction section to improve the reader's convinience.
- We have added information about the synthesis of peptides to the materials and methods section.
- We have added references to reviews and articles to the Introduction section.
Round 2
Reviewer 1 Report
Authors have substantially refined the article and made recommended edits. However, authors did not answer on my first question "What criterion was used to confirm that the data belongs to the normal distribution"? and did not make appropriate additions to the description of statistical data processing. I kindly ask dear authors to provide the answer and make this addition.
Author Response
We are thankful to the reviewer for providing very useful comments to our work. We have added information about statistical analysis in the Materials and Methods chapter. We have used Shapiro–Wilk test to confirm the data normality.